# Characterization of Trehalose-6-phosphate Synthase and Trehalose-6-phosphate Phosphatase Genes and Analysis of its Differential Expression in Maize (*Zea mays*) Seedlings under Drought Stress

**DOI:** 10.3390/plants9030315

**Published:** 2020-03-03

**Authors:** Phamela Acosta-Pérez, Bianka Dianey Camacho-Zamora, Edward A. Espinoza-Sánchez, Guadalupe Gutiérrez-Soto, Francisco Zavala-García, María Jazmín Abraham-Juárez, Sugey Ramona Sinagawa-García

**Affiliations:** 1Universidad Autónoma de Nuevo León, Facultad de Agronomía, Lab. de Biotecnología, Calle Francisco I. Madero S/N, Hacienda el Canadá, Cd Gral. Escobedo 66050, N.L., Mexico; phamela.acosta@gmail.com (P.A.-P.); juanita.gutierrezst@uanl.edu.mx (G.G.-S.); francisco.zavalagr@uanl.edu.mx (F.Z.-G.); 2Unidad de Genómica, Centro de Investigación y Desarrollo en Ciencias de la Salud, Campus Ciencias de la Salud, UANL, Dr. Carlos Canseco s/n, Mitras Centro, Monterrey 64460, N.L, Mexico; bdcz94@gmail.com; 3Universidad Autónoma de Chihuahua, Facultad de Ciencias Químicas, Circuito Universitario S/N, Campus Uach II, Chihuahua 31125, Chih, Mexico; eaespinoza@uach.mx; 4División de Biología Molecular, Instituto Potosino de Investigación Científica y Tecnológica, Camino a la Presa San José, Lomas 4 sección, San Luis Potosí 78216, S.L.P., Mexico; maria.abraham@ipicyt.edu.mx

**Keywords:** trehalose precursors, drought stress, *Zea mays*, osmotic adjustment

## Abstract

Maize is the most important crop around the world and it is highly sensitive to abiotic stress caused by drought, excessive salinity, and extreme temperature. In plants, trehalose has been widely studied for its role in plant adaptation to different abiotic stresses such as drought, high and low temperature, and osmotic stress. Thus, the aim of this work was to clone and characterize at molecular level the trehalose-6-phosphate synthase (TPS) and trehalose-6-phosphate phosphatase (TPP) genes from maize and to evaluate its differential expression in maize seedlings under drought stress. To carry out this, resistant and susceptible maize lines were subjected to drought stress during 72 h. Two full-length cDNAs of TPS and one of TPP were cloned and sequenced. Then, TPS and TPP amino acid sequences were aligned with their homologs from different species, showing highly conserved domains and the same catalytic sites. Relative expression of both genes was evaluated by RT-qPCR at different time points. The expression pattern showed significant induction after 0.5 h in resistant lines and after two to four hours in susceptible plants, showing their participation in drought stress response.

## 1. Introduction

Temperature fluctuations caused by climate change negatively affect most plants, causing chlorosis, slowing growth, and in extreme cases, plant death; in response to these conditions plants are capable of accumulate various organic compounds, such as soluble sugar and free amino acids known as osmoprotectants. One of these compounds is a sugar named trehalose that can accumulate in amounts up to 12% of dry weight in the cell to keep their integrity [1,2]. The trehalose is an osmolyte which is associated with tolerance to different kind of abiotic stresses, such as drought, high and low temperature, and osmotic stress, playing a vital role as mediator [3,4]. Trehalose is a natural disaccharide formed by two molecules of glucose bound by an α,α-1,1-glucosidic linkage. This carbohydrate is synthesized in many organisms, including bacteria, yeast, fungi, plants, insects, invertebrates, green weed, and cyanobacteria [4,5,6,7].

Physicochemical properties, such as the absence of reducing ends which are involved in forming glycosidic bond makes this disaccharide resistant to heat, pH and Maillard’s reaction, as well being able to form a glass-like structure that can stabilize enzymes, proteins, and lipids in membranes; this confers to trehalose not only a role as an energy source but also the capacity of protect biological structures from damage during abiotic stress [8,9,10]. Another advantage of this sugar is to be a signaling and antioxidant molecule; for example, trehalose-6-phosphate (T6P) plays a central role in growth, development and flowering in plants; likewise, the T6P regulates carbohydrate metabolism by controlling the entry of glucose into the glycolysis process [4,5]. Another report by Bae et al. [11] showed that trehalose can act as an elicitor of genes involved in detoxification and stress response, due to its ability to alter expression level of transcription factors, cell wall modification, nitrogen metabolism and fatty acid biosynthesis genes when it is applied in a exogenous way.

At least five trehalose biosynthetic pathways have been reported in different organisms [12]. The best-characterized pathway in the synthesis of trehalose involves two enzymes: trehalose-6-phosphate synthase (TPS) and trehalose-6-phosphate phosphatase (TPP). In the first step, TPS transfers glucose from UDP-glucose (UDPG) to glucose-6-phosphate (G6P) to generate trehalose-6-phosphate (T6P), which is dephosphorylated by TPP to produce free trehalose in the second step [12,13] (Scheme 1).

Moreover than being a pathway to produce trehalose, in higher plants, these enzymes have been well documented to act as signaling molecules that modulate an important number of metabolic and development processes in plants. For example, the enzyme TPS plays an important role in starch synthesis through the post-translational redox activity of ADP-glucose pyrophosphorylase, and in the case of TPP enzyme, it can inhibit the activity of Sn1-related protein kinase (SnRK1), which is known to play a main role in transcriptional networks in stress conditions and energy metabolism [14]. Similarly, Satoh-Nagasawa et al. [15] mentioned that trehalose may act as a signal in a specific developmental pathway, because they showed that a functional TPP enzyme can act upstream of the *RA1* transcription factor to regulate inflorescence branching. Different investigations have been carried out to study the overexpression of trehalose genes in plants to increase tolerance to abiotic stress like in *Arabidopsis thaliana*, where a significant increase in the drought tolerance, salinity, freezing, and heat stress was observed [16,17]. Also, Garg et al. [18] cloned the trehalose biosynthetic genes *ots*A (TPS), and *ots*B (TPP) from *E. coli*, followed by their expression in rice (*Oryza sativa* subsp. Indica var. PB-1) finding sustained plant growth, less photo-oxidative damage, and more favorable mineral balance under salt, drought, and low-temperature stress conditions in transgenic lines.

Additionally, Wang et al. [19] showed the expression of a functional trehalose-6-phosphate phosphatase gene in tobacco (*Nicotiana tabacum* L. var. Xanthi) under heat stress. In contrast, in *Zea mays* (inbred line 18-599), seaweeds (*Porphyra yezoensis, Porphyra haitanensis, Laminaria japónica, Undaria pinnatifid, Gracilaria lemaneiformis, Sargassum henslowianum, Monostroma angicava*, *Ulva pertusa*, *Chondrus ocellatus*, and *Enteromorpha prolifera*), trees (*Ginkgo biloba*) and mushrooms (*Pleurotus tuoliensis*), it was possible to clone and characterize trehalose-6-phosphate synthase and to show the role of this gene in resistance to abiotic stress [20,21,22,23]. However, many of the functions of TPS/TPP genes are unknown, especially those involved in signaling pathways in plant development and stress tolerance. Therefore, characterization of TPS and TPP genes is crucial for investigating the molecular mechanisms not only for increase our knowledge and understanding about abiotic stress strategies, but also to improve crop stress tolerance by gene manipulation [10,14].

Maize is one of the main crops around the world, not only for its importance in both human and animal consumption but also for its industrial uses [24]. This crop is highly sensitive to abiotic stress caused by drought, excessive salinity, and extreme temperature, reducing yield up to 15%, reaching an estimated loss of 16 million tons of grain, which makes necessary to implement strategies that minimize the losses caused by stress in this crop. Because trehalose is one of the osmolytes that, together with other compounds, could help to stabilize proteins and cellular membranes, and therefore can help in mitigation of abiotic stress damages, this study aimed to clone and characterize at molecular level the TPS and TPP genes in maize seedlings under drought stress.

## 2. Results

### 2.1. Cloning and Characterization of TPS and TPP cDNA

PCR analyses exhibited two fragments from TPS amplification, one of 2006 pb named TPS-2 gene and a 2625 bp fragment named TPS-3 gene. For TPP, an amplicon of 1145 bp was produced, which was named TPP-1. PCR fragments were cloned into pGEM-T Easy, and the plasmids were named as pAP2 and pAP3 for the isoforms of TPS-2 and TPS-3 and pAP5 for the TPP-1 variant, respectively. Finally, all plasmids were sequenced, and sequences were analyzed using BLAST (https://blast.ncbi.nlm.nih.gov/Blast.cgi).

### 2.2. Amino Acid Sequence Analysis and Phylogenetic Relationship

Sequence analysis showed an Open Reading Frame (ORF) that encodes a 608 amino acid protein with a predicted molecular weight of 76.23 kDa for TPS-2, whereas TPS-3 presented an ORF of 874 amino acids with a putative molecular weight of 97.39 kDa. Analysis of amino acid sequences revealed that these two proteins (TPS-2 and TPS-3) have two domains that correspond to the families of glycosyltransferase 20 and GT20_TPS. Glycosyltransferase is related to catalysis and subsequent synthesis of alpha, alpha-1,1-trehalose-6-phosphate, by using UDP-glucose as a donor (Figure 1a,b). Comparison of amino acid sequences between TPS-2 and TPS-3 showed a similarity of 53.5%; and, a computational modeling of both proteins was made, it showed a similarity of 92% between the two secondary structures and a DRMS of 9.22 Å among both (Figure 1c,d).

A multiple amino acids sequences alignment was made with those TPS reported in other organisms. TPS-2 has an identity of 89% to AtTPS (*Arabidopsis thaliana*, NP_172129.1) sequence, 89% to OsTPS (*Oryza sativa*, AEB53178.1), 88% to SbTPS (*Sorghum bicolor*, XP_021308880.1), and 85% to the one described in ZmTPS (*Zea mays*, B4FVF6). Meanwhile, TPS-3 showed 92% of identity to AtTPS, 91% to SbTPS, 92% to OsTPS, and 88% to the *Zea mays* (ZmTPS) reported. Also, a phylogenetic analysis was carried out; it highlighted that TPS-3 has a more recent evolutionary similarity compared to TPS-2 and TPS from other species such as sorghum (SbTPS), *Arabidopsis* (AtTPS) and rice (OsTPS) (Figure 1e).

For TPP-1, it was found an ORF encoding a 381 amino acids protein, with a theoretical molecular weight of 42.21 kDa, and it presented a trehalose-PPase domain. Also, computational modeling of this protein sequence was performed (Figure 2a,b). Concerning the comparison to different TPP sequences from other species, it was found a 82% similarity to the one described in *Zea mays* (ZmTPP, B4FVF6), 81% similarity to the one reported in *Oryza sativa* (OsTPP, Q9FWQ2), 80% to *Triticum urartu* (TuTPP, EMS61117.1) and 78% to the one in *Arabidopsis thaliana* (AtTPP, Q9SU39). Phylogenetic analysis showed that TPP-1, ZmTPP, and OsTPP have a closer relationship and a structural similarity unlike other species, so they probably derive from the same ancestor (Figure 2c).

### 2.3. In Silico Analysis of Catalytic Sites

*In silico* analysis of amino acid sequences revealed that these two proteins (TPS-2 and TPS-3) share the same structure and location of the active site. Highly conserved residues were found, trp^40^, tyr^76^, trp^85^ and arg^300^, which are associated to the binding sites with glucose-6-phosphate and asp^130^, his^154^, arg^262^, and asp^361^ amino acids, which are part of the binding sites to UDP- glucose and amino acids in the catalytic site his^154^ and asp^36^ (Figure 3a,b). Likewise, both TPS-2 and TPS-3 showed 92% and 93% of similarity, respectively, to other TPS sequences like AtTPS, SbTPS, OsTPS, ScTPS (*Saccharomyces cerevisiae*, NP_009684.1), CaTPS (*Candida albicans*, XP_711706.1), ZmTPS and EcTPS (*Escherichia coli*, P31677) (Figure 3c).

On the other hand, TPP-1 was 80% similar to other TPPs from different organisms previously reported, like TaTPP, ZmTPP, OsTPP; AtTPP, ScTPP (*Saccharomyces cerevisiae*, CAA50025.1), CaTPP (*Candida albicans*, XP_721536.1), and EcTPP (*Escherichia coli*, P31678). For TPP-1 amino acid sequence, the presence of the most conserved residues: asp^20^, asp^22^, and asp^198^, which are related to the substrate-binding sites and catalytic site was identified. Asp^20^, leu^21^ and asp^22^ correspond to the binding sites to the substrate and asp^20^ to the active site. (Figure 4a,b).

### 2.4. Real-Time Quantitative PCR

Relative expression level of TPS-3 and TPP-1 in resistant and susceptible plants in response to drought stress was determined. In the case of expression of the TPS-3 gene, resistant seedlings significantly increased during the first 30 min and again two hours later to decreased to normal levels from four hours until 72 h. In susceptible seedlings on the other hand, there was an upregulation in the expression of TPS-3 from two hours to six hours, and then a downregulation of the expression to the 12 h after start of stress conditions.

Finally, this gene was upregulated from 24 h to 60 h in stress (Figure 5a). In the expression of the TPP-1 in resistant plants, there was a high induction in the first hour, and the expression levels decreased to 12 h after drought stress and at 18 h it was upregulated continuously. Meanwhile, TPP-1 gene in susceptible seedlings was upregulated from the first hour until the fourth hour and downregulated to 12 h after start the stress treatment. Then, the expression level increased from 18 h to the end of the experiment (Figure 5b). Statistical analysis showed a high correlation between the two main factors: genotype and gene expression (p ≤ 0.05) in both cases during the time of the experiment, founding that the expression of these genes could be related to the genotype. It was observed that in the case of TPS-3 expression there was statistical difference between the lines used at 0.5, 4, 6, 8, 10, 12, 72 h, been the resistant genotype the one that shown more upregulared behavior compared with the susceptible plants. In the same way, TPP-1 gene expression showed a significant difference between genotypes at 0.5, 1, 2, 4, 8, 10, 12, 18, 48 and 60 h. Data showed that the resistant line had higher expression level in these hours unlike what was observed in susceptible plants.

## 3. Discussion

Water stress, as well as other kind of abiotic stress, affects plant behavior in different ways: genetic, transcriptomic, proteomic, metabolic and physiological. In plants, growth is hardly affected at low-stress levels. In maize, abiotic stress inhibits roots and leaves growth in both moderate and high stress level [25]. Similarly, the number and size of leaves and plant height are affected in maize plants under drought stress much more than root growth [26,27].

The osmolyte compatible trehalose has shown the ability to protect organisms under drought stress in several species. The main biosynthetic pathway TPS-TPP has been studied extensively. In this study, two TPS and one TPP genes were identified in *Zea mays*. The cDNAs cloned from maize showed that the *TPS-2* and *TPS-3* contained Glyco_transf_20 and GT20_TPS domains, and the *TPP-1* gene showed a Trehalose_PPase domain. These results were similar to the ones showed by Wang et al. [14], where the *Bd*TPS and *Bd*TPP genes showed these type of domains, which are related to the activity of theses enzymes. On the other hand, amino acid sequence analysis showed high homology comparing to other organisms. Alignment of TPS and TPP protein sequences from bacteria, yeast, and plants showed highly conserved residues involved in substrate binding and catalysis [21,28,29,30]. Most conserved domains of these proteins are involved in the enzyme-substrate binding and the catalytic site in the synthesis of trehalose-6-phosphate. Even though the tertiary structure of both proteins, *TPS-2* and *TPS-3*, is different visually, they share the same catalytic sites. This result agrees with what was reported by Gibson et al. [28] and Jiang et al. [21], who observed that all known plant functional TPS proteins show conservation in these amino acids.

The relative expression of both genes, *TPS-3* and *TPP-1*, was induced after stress, this behavior could be related to their role in drought stress tolerance, which is a common characteristic of TPS and TPP in different organisms. Recently, trehalose metabolism has been related to stress tolerance and some experiments have indicated the role of TPS genes in increasing stress tolerance [31]. Respect to the expression of TPS genes, these results were similar to those reported by Jiang et al. [21], where expression of TPS in maize was upregulated in the first hours of drought stress. Likewise, the results showed by Wang et al. [14] where the expression levels of some TPS genes isolated increased under different types of abiotic stress like chilling, salinity and drought stress in the first three hours. Also, the author mention that this expression patterns can be connected with various aspects of abiotic stress tolerances. In the same way, this behavior may be due to Ca^+^ influx into the cell cytoplasm, which functions as an intracellular second messenger molecule, that stimulates accumulation of compatible osmolytes like trehalose [20,32,33]. In addition, to the requirement of trehalose accumulation and maintaining of gene expression, to produce this osmoprotectant in low levels [34]. Another study about TPS genes in Arabidopsis [35] suggests an interaction between TPS expression and ABA metabolism and its role as the second messenger, which can explain rapid upregulation in abiotic stress.

TPP expression showed upregulation in the first two hours, with a faster response in resistant plants, unlike susceptible ones. This response was similar to the one reported by Ge et al. [34], who observed upregulation of the rice TPP at the first hour in stress conditions, as well as, the relative expression level of *BdTPPC* gene that was upregulated in the first hour under abiotic stress [14]. Overexpression of TPP has been documented in flowering stages in maize and has been shown its relationship to yield improvement during drought stress [36], furthermore, Satoh-Nagasawa et al. [15] mentioned that trehalose may act as a signal in a specific development pathway, because they showed that a functional TPP enzyme can act like transcription factor to regulate inflorescence branching. In this study, it was observed that resistant plants showed a faster response than susceptible plants, suggesting the participation of these genes in drought tolerance. In addition, because TPS/TPP can respond to ABA, and ANA-responsive elements (ABRE), it may be suggested that TPS and TPP genes are involved in dehydration response by ABA signaling pathway [14].

## 4. Materials and Methods

### 4.1. Plant Material, Growth Conditions and Stress Treatment

Two maize lines were used to conduct this work, (CML 311) and (CML 551), which were catalogued like susceptible (S) and resistant (R), respectively, to drought, high temperature, and low level of nitrogen by the International Maize and Wheat Improvement Center (CIMMYT, Texcoco México) whom supplied it. Seeds were germinated in a greenhouse in a plastic pot (15 × 15″). Substrate mix was perlite, pumice, and peat moss to 1:1:3 ratio. Plants were kept at 12 h of photoperiod, 61% relative humidity, and 30 °C.

### 4.2. Watering Treatments

The following watering treatments were applied to three-leaf stage maize plants: (D) a drought stress treatment consisting of maintaining the plants without watering until 15–20% of field capacity and (W) a well-watered treatment with 100% of field capacity. The moisture monitoring was carried out by weighing the pots daily during 72 h. A pool of twelve leaves was harvested at 0.5, 1, 2, 4, 6, 8, 10, 12, 18, 24, 48, 60 and 72 h after starting the drought treatment (RD and SD) and frozen on dry ice and stored at −70^◦^°C. Control seedlings (RW and SW) were collected at the same time. All the treatments were conducted in a completely randomized 2 × 2 factorial design with 12 replications.

### 4.3. RNA Extraction and cDNA Synthesis

Total RNA was extracted from *Z. mays* plants under drought stress using the protocol reported by Stiekema et al. [37]. A cDNA library was designed through reverse transcription (RT) using the High-Capacity cDNA Reverse Transcription Kit (Applied Biosystems, Carlsbad, CA, USA) according to the manufacturer’s instructions. The first-strand cDNA was used as a template in PCR and qPCR amplification.

### 4.4. Cloning TPS/TPP cDNA and Sequence Analysis

TPS and TPP coding sequences were obtained by PCR using the cDNA product from the RT reaction. Search for information in the NCBI showed several predicted and putative sequences. On that, two sequences were used to amplify the TPS gene: the putative sequence NC_024460.2 from the TPS4 gene (*TPS-2*) and the sequence GU228585.1 from TPS2 gene (*TPS-3*). Whereas, for *TPP-1*, the sequence NM_001158750.1 was used.

Based on predicted maize TPS/TPP genes, primers were designed and used to amplify the TPS and TPP genes from maize cDNA. For the *TPS-2* gene: F-ATGGTTCTGAAGTCGCACACA and R-TCAGCTGCTTTGTTCCATCTGA were used, while for the *TPS-3* gene: F-ATGTCGCGGG TGATGACG and R-CTAAGACCCTCCAATTGGTGT were used, amplifying a product of 2.0 and 2.6 kb, respectively. In the case of the *TPP-1* gene the following primers were used: F-ATGGATTTGAAGACAGGCCTC and R-TCAGGTGGACTGCTCCTTC with a product of 1 kb approximately. PCR was made using the Platinum Taq DNA Polymerase kit (Invitrogen, Waltham, MA, USA), and the amplified products were purified and cloned into pGEM-T Easy (pGEM^®^-T and pGEM^®^-T Easy Vector Systems (PROMEGA, Madison, WI, USA) according to the manufacturer’s instructions and sequenced by external service.

After sequencing and assembling, sequence analysis was made. A homology search was performed using BLASTX from the NCBI database. Sequences of TPS and TPP genes that showed similarity to the cloned ones were used for a full amino acid sequence alignment using CLUSTALX and Coffemate for in silico analysis. The amino acid sequences were used for protein modeling using Phyre2; also, the structures were aligned using Matras software and visualized in PyMOL, and phylogenetic analysis was carried out using the Maximum-Likelihood algorithm using MEGA 4.0

### 4.5. Real-Time Quantitative PCR

qPCR was performed on LightCycler 480 using PowerUP^TM^ SYBR^TM^ Green Master Mix (Applied Biosystems) according to the manufacturer’s instructions. To determine gene expression, a pair of primers for *TPS-3* (F- AAGTCAAACCACAGGGAGTAAG/R-TGTCCTCATCGGACCTATCAT) and *TPP-1* (F- TGGAGATGACAGAACAGATGAAG/R- GCGTCACTCTCTTTGGGTATAG) and *18S* ribosomal RNA gene (F- CTGAGAAACGGCTACCACA/R-CCCAAGGTCCAACTACGAG) were used as endogenous control. The temperature protocol was one cycle of 120 s at 50 °C and other cycle of 120s at 95 °C, 45 cycles of 15 s at 95 °C, 15s at 60 °C, and 60 s at 72 °C, fluorescence was detected at 72 °C. The relative fold change in gene expression was calculated using the 2 ^(−ΔΔCt)^ method, where, ΔCt = (Ct target gene—Ct reference gene), then the ΔCts were normalized using the Endogenous Control gene ΔΔCt = (ΔCt treated—ΔCt untreated) and finally the differential expression the TPS and TPP genes were calculated and represented in Log 2 scale.

### 4.6. Statistical Analysis

The qPCR results were reported as Log_2_ 2^(−ΔΔCt)^. The statistically significant difference between the expression level was analyzed with a one-way ANOVA, and treatment means were compared with a Tukey Test (*p* ≤ 0.05) using Excel 2016 (Microsoft, Redmond, WA, USA) and SPSS Statistical Package version 25 (SPSS Inc., Chicago, IL, USA).

## 5. Conclusions

Currently, climate change negatively affects many crop plants, which accumulate different compounds to maintain protein integrity in their cells. In this sense, understanding the mechanisms that govern plant adaptation to stress will allow us to design strategies to improve crop yield. In this research, we cloned the TPS and TPP genes from maize seedlings and found that these genes are differentially expressed under drought stress. Corresponding analysis showed that maize has different intermediary enzymes for trehalose production, which share a high degree of conservation of residues present in their catalytic sites, and these are similar to functional TPS and TPP from other organisms. This study presents information about the different TPS and TPP isoforms expressed in maize, however it is necessary to conduct further research on the participation of these genes and other compounds involved in signaling pathways that activate different molecular mechanisms involved in conferring tolerance to abiotic stress in plants.

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
