# Peer review of "Characterization of Trehalose-6-phosphate Synthase and Trehalose-6-phosphate Phosphatase Genes and Analysis of its Differential Expression in Maize (Zea mays) Seedlings under Drought Stress"

_plants, 2020, doi:10.3390/plants9030315_

Round 1
Reviewer 1 Report
The authors have revised the manuscript well I recommend publication of the revised version.
Author Response
The MS suffered some important changes based on the suggestions of the academic’s editor in chief. For example, in the general objective we only considered the cloning, in silico analysis and gene expression analysis. The physiologic and the biochemical analysis such as content of chlorophyll, photosynthetic pigments, proline, and the SOD activity were removed, because we think these can be complemented with other data we are working on and can be published separately.
Given these modifications the MS suffered changes in the abstract, introduction, results, discussion, and conclusions. Lastly, references were updated.
Also, it was not possible to answer some of the reviewer’s comments due to the changes in the MS. Please find all these modifications highlighted on the MS.
English was reviewed by a native speaker.
Reviewer 2 Report
The MS was improved but some previous responses are not correct and still several points need to be corrected /modified.
Responses to reviewer 2:
In the present form the MS fails to demonstrate that maize TPS and TPP have a role in plant stress response because Fig. 6 is not correctly presented; moreover, I remain convinced that the mere characterization of two genes it is not a condition for the publication of a scientific article. In fact, the scientific articles presented in the response linked the gene cloning to figures or tables showing in addition some physiological effects. Anyway, I appreciate the combativeness of the authors and will present below indications to further improve MS. The abbreviations for the treatments/controls remain difficult to understand; why not indicate with R the resistant plants, with S the susceptible plants, with D the plants subjected to controlled drought stress, and with W the well-watered plants? In this way, the 4 abbreviations would be immediately understandable: RD, RW, SD, SW. Leaves number, height plant and DW of leaves / seedlings should be reported as a table in supplementary materials.Further comments:
Because in the abstract (line 27) it is indicated that photosynthetic pigments decreased, the corresponding values should be reported as a Table or a Figure in supplementary materials. In abstract, (line 31-32) the phrase “This study presented basic information on the different TPS and TPP isoforms expressed in maize” corresponds to an insignificant result; please correct in “ The relative expression of both genes, TPS and TPP, was induced significantly after stress, showing their participation in responding to drought stress” as written at lines 315-316 or something with a similar meaning. 1 and Fig. 6 should show a similar sample timing; if Fig. 1 presents only data at 72h that means a major relevance of that timing so that Fig. 6 needs to be redrawn giving a space directly proportional to the time / hours of analysis in the x axis to show that for over 50 hours (after 12-16 h) the expression levels remain for TPS and TPP in susceptible plants and about constant / near zero values in resistant plants. Obviously, this should be discussed in the text. 1B: the part below zero must be omitted.Author Response
The MS suffered some important changes based on the suggestions of the academic’s editor in chief. For example, in the general objective we only considered the cloning, in silico analysis and gene expression analysis. The physiologic and the biochemical analysis such as content of chlorophyll, photosynthetic pigments, proline, and the SOD activity were removed, because we think these can be complemented with other data we are working on and can be published separately.
Given these modifications the MS suffered changes in the abstract, introduction, results, discussion, and conclusions. Lastly, references were updated.
Also, it was not possible to answer some of the reviewer’s comments due to the changes in the MS. Please find all these modifications highlighted on the MS.
English was reviewed by a native speaker.
The answers to reviewers’ comments are the following:
Reviewer 2:
In the present form, the MS fails to demonstrate that maize TPS and TPP have a role in plant stress response because Fig. 6 is not correctly presented;
Following the suggestion of the academic’s editor in chief and the reviewers, the MS has had several changes. One of them was the objective that relates the expression of these genes in response to drought stress.
moreover, I remain convinced that the mere characterization of two genes it is not a condition for the publication of a scientific article.
The referent of the mere characterization of genes is not a condition for publication of a scientific article; the articles demonstrated bellow focus their studies in the isolation and characterization of these genes in different organisms.
Wu, Weisheng, Yongzhen Pang, Guo-An Shen, Jie Lu, Juan Lin, Jin Wang, Xiaofen Sun, y Kexuan Tang. “Molecular Cloning, Characterization and Expression of a Novel Trehalose-6-Phosphate Synthase Homologue from Ginkgo Biloba”. BMB Reports 39, núm. 2 (el 31 de marzo de 2006): 158–66. https://doi.org/10.5483/BMBRep.2006.39.2.158.
Wang, Guoliang, Ge Zhao, Yanbin Feng, JinsongXuan, JianweiSun, BaotaiGuo, GuoyongJiang, et al. “Cloning and Comparative Studies of Seaweed Trehalose-6-Phosphate Synthase Genes”. Marine Drugs 8, núm. 7 (el 6 de julio de 2010): 2065–79. https://doi.org/10.3390/md8072065.
Wang, Song, Kai Ouyang, y Kai Wang. “Genome-Wide Identification, Evolution, and Expression Analysis of TPS and TPP Gene Families in BrachypodiumDistachyon”. Plants 8, núm. 10 (el 23 de septiembre de 2019): 362. https://doi.org/10.3390/plants8100362.
Abbreviations for the treatments/controls remain difficult to understand; why not indicate with R the resistant plants, with S the susceptible plants, with D the plants subjected to controlled drought stress, and with W the well-watered plants? In this way, the 4 abbreviations would be immediately understandable: RD, RW, SD, SW.
Following the suggestion of the reviewer, all labels were changed for a better understanding of the abbreviations of the treatments. See Materials and Methods (Watering treatments), lines 1588
Leaves number, height plant and DW of leaves / seedlings should be reported as a table in supplementary materials.
Given the changes in the MS, this part was removed from the main text.
Further comments:
Because in the abstract (line 27) it is indicated that photosynthetic pigments decreased, the corresponding values should be reported as a Table or a Figure in supplementary materials.
Given the changes in the MS, this part was removed from the main text.
In abstract, (line 31-32) the phrase “This study presented basic information on the different TPS and TPP isoforms expressed in maize” corresponds to an insignificant result; please correct in “ The relative expression of both genes, TPS and TPP, was induced significantly after stress, showing their participation in responding to drought stress” as written at lines 315-316 or something with a similar meaning.
Following the suggestion of the reviewer, this sentence was modified for a better understanding of the participation of these genes in response to drought stress. See line: 29-31
Fig 1 and Fig. 6 should show a similar sample timing; if Fig. 1 presents only data at 72h that means a major relevance of that timing so that Fig. 6 needs to be redrawn giving a space directly proportional to the time / hours of analysis in the x axis to show that for over 50 hours (after 12-16 h) the expression levels remain for TPS and TPP in susceptible plants and about constant / near zero values in resistant plants.
Given the changes of the MS, the part corresponding to the results showed in figure 1 was removed from the main text. Thus, data presented in figure 6 were kept since this information corresponds to the second part of the main objective of the study that evaluated the expression of the TPS and TPP in the first 72 hours under drought stress.Obviously, this should be discussed in the text. 1B: the part below zero must be omitted.
Given the changes in the MS, this part was removed and cannot be added to the discussion.
Reviewer 3 Report
While I appreciate the effort of the work presented, I recommend that this paper should be submitted to other journals.
Overall, extensive editing of English language and style is required. For example, the last sentence in Abstract is uncomfortable.
In Figures, the authors should not use the letters that the readers cannot understand. Please ensure the figures have the appropriate resolution.
I hope the additional experiments that can support the biological significance of trehalose in maize under drought conditions.
Author Response
The MS suffered some important changes based on the suggestions of the academic’s editor in chief. For example, in the general objective we only considered the cloning, in silico analysis and gene expression analysis. The physiologic and the biochemical analysis such as content of chlorophyll, photosynthetic pigments, proline, and the SOD activity were removed, because we think these can be complemented with other data we are working on and can be published separately.
Given these modifications the MS suffered changes in the abstract, introduction, results, discussion, and conclusions. Lastly, references were updated.
Also, it was not possible to answer some of the reviewer’s comments due to the changes in the MS. Please find all these modifications highlighted on the MS.
English was reviewed by a native speaker.
The answers to reviewers’ comments are the following:
Replies to reviewer 3:
While I appreciate the effort of the work presented, I recommend that this paper should be submitted to other journals.
In this case, another article with the same theme has been published recently in this journal without any problem. The reference can be seen below:Wang, Song, Kai Ouyang, y Kai Wang. “Genome-Wide Identification, Evolution, and Expression Analysis of TPS and TPP Gene Families in Brachypodium distachyon.”Plants 8, no.. 10 (September 23rd,2019): 362. https://doi.org/10.3390/plants8100362.
Overall, extensive editing of English language and style is required. For example, the last sentence in Abstract is uncomfortable.
The English language was reviewed by a native speaker.In Figures, the authors should not use the letters that the readers cannot understand. Please ensure the figures have the appropriate resolution.
All figures were replaced for better resolution and understanding of the data shown.I hope the additional experiments that can support the biological significance of trehalose in maize under drought conditions.
In the case of this MS, the goal was to clone and characterize at molecular level the three genes under study and evaluate its expression pattern shown by maize seedlings under drought stress. That is why our work focused mostly on analyzing the expression of these genes and their molecular behavior in the first 72 hours.
Reviewer 4 Report
The authors present the Characterization of two enzymes involved in thehalose pathways, at the gene levels. They perform an experiment with two maize lines known to be resistant and susceptible to drought, and they tried to correlate the expression levels of the genes coding for these two enzymes with the ability of the line to resist to drought.
Major concern :
The is no statistical analysis for qRT-PCR data, which is absolutely missing. Please add this information, so you will be able to compare the effect of the treatment in the two maize lines, not only in terms of dynamics but also in terms of quantification.Other comments:
+ Abstract:
Line 24 : please define TPS and TPP
+ Introduction:
Line 41: correct the typo “different kind of abiotic stress” should be changes into “different kind of abiotic stresses” Lines 45-50 : I suggest to illustrate the pathways involved in the synthesis of trehalose with a figure Lines 51-54 : This sentence is not clear, because you mention that enzymes act as signaling molecule (which is not true), and you illustrate with an example with trehalose-6-phosphate that is not an enzyme. So please be more precise. Lines 52-53: the sentence is not correct. Change into “signaling molecule that modulate an important number of metabolic and development processes”.
+ Results:
Figure 1a: I would rather see ORF from TPS-2 and TPS-3 separately on the figure. Figure 1b and 1c : please homogenize the legends Figure 5: the legend is not precise enough. You should mention that the expression of the genes is expressed relatively to the well-watered conditions. Figure 5: Show the data about TPS2 otherwise, we don’t understand why it was presented before in the article. The maize lines that have been used are described as “resistant” and “susceptible”, but there is no result to show that it is the case in the study. Could you prove it?
+ Material and Methods:
Lines 252-253: the maize lines that have been used are described as “resistant” and “susceptible”, but there is no reference to prove it. Please add the references. Lines 259-261: The drought treatment is not well explained as it is not the same unit that is used to explain the watering conditions in D and W plants. For D treatment, it is expressed as a % of field capacity while for W it is expressed as a % irrigation The time to reach the 15-20% field capacity is not explained, which is a problem when the harvests are so precisely defined (0.5, 1, 2… h after starting the drought treatment). The homogeneity of the plants should be at least precised, concomitantly with the %field capacity at the moment of the harvest, and used as a covariable in the statistics of qRT-PCR.
+ Discussion:
Conclusions about the relationship between the expression pattern of TPS and TPP gene expression and the ability of the plant to tolerate drought should be moderated because :1- there is no evidence of such relationship in the present paper:
just correlations! and it is hard to be convinced that a 1 hour decay in the up regulation of gene expression can explain the resistant/susceptible phenotype. You should discuss on the level of the up-/down- regulation of TPS3 and TPP1 after drought.2-only two genotypes were studied.
Author Response
Dear Ms. Hinata Fang
Assistant Editor Plants
Thank you for your attention for the revision of the manuscript entitled “Characterization of trehalose-6-phosphate synthase and trehalose-6-phosphate phosphatase genes and analysis of its differential expression in maize (Zea mays) seedlings under drought stress, (Manuscript ID: plants-701585) please find attached the revised version where we have attended all comments and suggestion of the reviewers in the manuscript.
The responses to reviewers’ comments are the following:
Academic editor comments
Even though the authors’ effort to respond to the reviewers’ comment and suggestion,
the revised manuscript has still several points to be addressed.
- The comments and suggestions of Reviewer 4 and 5 should be clearly addressed.
Response: The comments of the reviewer 4 and 5 were attended. Please see below
- Full discussion of the present data with two given references-below to advocate the novelty and significance of the present manuscript.
1. Wang, Song, Kai Ouyang, y Kai Wang. “Genome-Wide Identification, Evolution, and Expression Analysis of TPS and TPP Gene Families in Brachypodium distachyon. ”Plants 8, no.. 10 (September 23, 2019). 362.
2. Nature. 2006 May 11;441(7090):227-30. “A trehalose metabolic enzyme controls inflorescence architecture in maize". .http://doi.org/10.3390/plants8100362, is a genome-wide study.
Response: These manuscripts were considered and cited into introduction and discussion.
Reviewer 4
Principio del formulario
Review Report Form
The authors present the Characterization of two enzymes involved in thehalose pathways, at the gene levels. They perform an experiment with two maize lines known to be resistant and susceptible to drought, and they tried to correlate the expression levels of the genes coding for these two enzymes with the ability of the line to resist to drought.
Major concern:
There is no statistical analysis for qRT-PCR data, which is absolutely missing. Please add this information, so you will be able to compare the effect of the treatment in the two maize lines, not only in terms of dynamics but also in terms of quantification.
Response: Response: Now Materials and Methods described how the data of qRT-PCR were reported: “The relative fold change in gene expression was calculated using the 2 (-ΔΔCt) method, where, ΔCt = (Ct target gene - Ct reference gene), then the ΔCts were normalized using the Endogenous Control gene ΔΔCt = (ΔCt treated - ΔCt untreated) and finally the differential expression the TPS and TPP genes were calculated and represented in Log 2 scale”. Please see lines 341-344.
Fig. 5 now changed to Fig 6. Statistical analysis was added; as well as, the data showed was normalized using Log2 2(-ΔΔCt), where the endogenous control gene (well-watered fold change) take the value of 0. Thus, the level expression present was the fold change of the gene of interest based in stress treatment. The qPCR results were reported as Log2 2(-ΔΔCt). The statistically significant difference between the expression level was analyzed with a one-way ANOVA, and treatment means were compared with a Tukey Test (p ≤ 0.05) using Excel 2016 and SPSS statistical package version 25. See lines 345-348
Other comments:
Reviewer 4:+ Abstract: Line 24 : please define TPS and TPP
Response: TPS and TPP were defined in abstract. Please see line 24 and 25.
Reviewer 4 + Introduction:
Line 41: correct the typo “different kind of abiotic stress” should be changes into “different kind of abiotic stresses”
Response: different kind of abiotic stress” was changes to “different kind of abiotic stresses” Please see lines 43.
Lines 45-50: I suggest to illustrate the pathways involved in the synthesis of trehalose with a figure.
Response: Ok, trehalose pathway figure was included, please see figure 1
Lines 51-54: This sentence is not clear, because you mention that enzymes act as signaling molecule (which is not true), and you illustrate with an example with trehalose-6-phosphate that is not an enzyme. So please be more precise.
Response: You are right, the sentence “trehalose-6-phosphate play a central role in growth, development and flowering in plants; likewise, the T6P regulates carbohydrate metabolism by controlling the entry of glucose into the glycolysis process” was eliminated of this paragraph, and the paragragraph was changed to:
“More than being a pathway to produce trehalose; in high plants, these enzymes have been well documented acting as signaling molecule that modulate an important number of metabolic and development processes in plants; for example, TPS enzyme play an important role in starch synthesis through the post-translational redox activity of ADP-glucose pyrophosphorylase, and in the case of TPP enzyme, it can inhibit the activity of Sn1-related protein kinase (SnRK1), which is known to play a vital role in transcription networks of plant stress and energy metabolis13….” Please see lines 65-73
Lines 52-53: the sentence is not correct. Change into “signaling molecule that modulate an important number of metabolic and development processes”.
Response: Ok the sentence was change by: signaling molecule that modulate an important number of metabolic and development processes. See line 66 and 67.
+ Results:
Figure 1a: I would rather see ORF from TPS-2 and TPS-3 separately on the figure.
Response: Ok, the figure 2a shown the domains of TPS-2 and the domains of TPS-3 was added separately. Please see figure 2b.
Figure 1b and 1c: please homogenize the legends:
Response: Ok the Figure legend was homogenized: Please see Figure 2…(c) Computational modeling of TPS-2; (d) Computational modeling of TPS-3.
Figure 5: the legend is not precise enough. You should mention that the expression of the genes is expressed relatively to the well-watered conditions.
Response: you are right. Please see legend figure 6.
Figure 5: Show the data about TPS2 otherwise, we don’t understand why it was presented before in the article.
Response: Ok you are right, the qRT-PCR analyses of TPS-2 was accomplished; however, the results were not added to the manuscript because there was absence of expression for this gene. This could be related to the low quantity of mRNA of the gene that could not exceed the detection threshold.
The maize lines that have been used are described as “resistant” and “susceptible”, but there is no result to show that it is the case in the study. Could you prove it?
Response: The lines CML 551 and 311 were provided by international center for the improvement of maize and wheat (CIMMYT) as line tolerant and susceptible to drought stress. Please see link https://data.cimmyt.org/dataset.xhtml?persistentId=hdl:11529/10246
And under personal communication with personal of CIMMYT, We only used for this study as lines tolerant and susceptible, but we don’t have previous studies to confirm the genotype.
+ Material and Methods:
Lines 252-253: the maize lines that have been used are described as “resistant” and “susceptible”, but there is no reference to prove it.
Response: see response above.
Please add the references. Lines 259-261: The drought treatment is not well explained as it is not the same unit that is used to explain the watering conditions in D and W plants. For D treatment, it is expressed as a % of field capacity while for W it is expressed as a % irrigation.
Response: ok you are right, we homologue using % field capacity: (D) a drought stress treatment consisting of maintaining the plants without watering until 15-20% of field capacity and (W) a well-watered treatment with 100% of field capacity, for 72 h. Please see lines 298-301.
The time to reach the 15-20% field capacity is not explained, which is a problem when the harvests are so precisely defined (0.5, 1, 2… h after starting the drought treatment).
Response: The % of field capacity was monitored daily based on the weight of the pot until all reached an interval of 15-20% of its capacity. Samples were collected when they placed within this range. Please see lines 301-302.
The homogeneity of the plants should be at least precised, concomitantly with the %field capacity at the moment of the harvest and used as a covariable in the statistics of qRT-PCR.
Response: Ok, in this case we considered the range 15-20% of field capacity and the weight of the pot at the time of the sample collection was not registered, so the percentage of moisture could not be calculated. Likewise, we appreciate the comment to take it in future test+ Discussion:
Conclusions about the relationship between the expression pattern of TPS and TPP gene expression and the ability of the plant to tolerate drought should be moderated because:
there is no evidence of such relationship in the present paper:
just correlations! and it is hard to be convinced that a 1 hour decay in the up regulation of gene expression can explain the resistant/susceptible phenotype. You should discuss on the level of the up-/down- regulation of TPS3 and TPP1 after drought.
only two genotypes were studied.
Response: Ok you are right, the discussion was modified based to your comments, please see discussion session.
Sincerely
Dra. Sugey Ramona Sinagawa-García
Corresponding author
Reviewer 5 Report
During the last decades we can see growing interest in research of plant drought tolerance, mitigation of drought and possibilities to increase soil fertility in arid agricultural area. Different approaches can diminish the negative drought effects with use of different tools, which can have an economic impact worldwide.
Trehalose is a natural disaccharide formed by two molecules of glucose and may preserve the stability of the chloroplast envelope and maintain the osmotic potential of the chloroplast. Interaction of salinity and Trehalose improved photosynthetic pigment levels.
Trehalose is one of the compatible solutes, which are highly soluble, low molecular weight compounds which are usually non-toxic at high cellular concentrations. Generally, such solutes protect plants from stress through different courses, including contribution to cellular osmotic adjustment, detoxification of reactive oxygen species, protection of membrane integrity, and stabilization of enzymes and proteins.
Philosophy of this work is very good, applicable in basic and applied plant sciences. The innovation of the paper is strongly convected to the journal. The manuscript contains original data. The article is well organized. Material, Methods, and Protocols are standard. The experimental design was appropriate and the applied protocols were used correctly. The figures, captations content are considerable, readable and useful. The results of the experiments provide novel original data.
Specific comments:
- Add new keywords – osmotic adjustment,
- What is the capacity for the accumulation of trehalose in cells
- Does trehalose play a protective role against photoinhibition and high temperature similarly than proline?
- Write more about the physiological significance of trehalose in plants. It was recently published about the improving effect of trehalose may be due to its physicochemical properties that stabilize dehydrated enzymes, proteins, and lipid membranes, as well as protect structures from damage during desiccation. Can trehalose improve the stability of antioxidative enzymes under stress?
- What is the role of trehalose in alleviating in improving crop tolerance to oxidative stress induced by drought and salinity?
- Add characteristics of water stress during the experiment (RWC or leaf water potential) if it is possible.
- Can be trehalose application recommended to farmers as an effective measure for the strengthen ecostability of physiological processes of plants under drought environments?
- Add more about osmotic adjustment. I would like to propose to improve introduction and discussion about the physiological significance of this kind of research, using some very important and innovative topical papers, e.g.:
* Abdelgawad ZA, Hathout TA, El-Khallal SM, Said EM, Al-Mokadem AZ.: Accumulation of trehalose mediates salt adaptation in rice seedlings. Am Eurasian J Agric Environ Sci 2014, 14(12):1450–1463
* Zhang D, Tong J, He X, et al.: A Novel Soybean Intrinsic Protein Gene, GmTIP2;3, Involved in Responding to Osmotic Stress. Front. Plant Sci. 2016, 6:1237. doi: 10.3389/fpls.2015.01237
* Mostafa MG, Hossain MA, Fujita M.: Trehalose pretreatment induces salt tolerance in rice (Oryza sativa L.) seedlings: oxidative damage and coinduction of antioxidant defense and glyoxalase systems. Protoplasma, 2015, 252: 461–475
* Ma C, Wang Z, Kong B, Lin T: Exogenous trehalose differentially modulate antioxidant defense system in wheat callus during water deficit and subsequent recovery. Plant Growth Regul., 2013, 70:275–285
Author Response
Dear Ms. Hinata Fang
Assistant Editor Plants
Thank you for your attention for the revision of the manuscript entitled “Characterization of trehalose-6-phosphate synthase and trehalose-6-phosphate phosphatase genes and analysis of its differential expression in maize (Zea mays) seedlings under drought stress, (Manuscript ID: plants-701585) please find attached the revised version where we have attended all comments and suggestion of the reviewers in the manuscript.
The responses to reviewers’ comments are the following:
Academic editor comments
Even though the authors’ effort to respond to the reviewers’ comment and suggestion,
the revised manuscript has still several points to be addressed.
- The comments and suggestions of Reviewer 4 and 5 should be clearly addressed.
Response: The comments of the reviewer 4 and 5 were attended. Please see below
- Full discussion of the present data with two given references-below to advocate the novelty and significance of the present manuscript.
1. Wang, Song, Kai Ouyang, y Kai Wang. “Genome-Wide Identification, Evolution, and Expression Analysis of TPS and TPP Gene Families in Brachypodium distachyon. ”Plants 8, no.. 10 (September 23, 2019). 362.
2. Nature. 2006 May 11;441(7090):227-30. “A trehalose metabolic enzyme controls inflorescence architecture in maize". .http://doi.org/10.3390/plants8100362, is a genome-wide study.
Response: These manuscripts were considered and cited into introduction and discussion.
Reviewer 5
Comments and Suggestions for Authors
During the last decades we can see growing interest in research of plant drought tolerance, mitigation of drought and possibilities to increase soil fertility in arid agricultural area. Different approaches can diminish the negative drought effects with use of different tools, which can have an economic impact worldwide.
Trehalose is a natural disaccharide formed by two molecules of glucose and may preserve the stability of the chloroplast envelope and maintain the osmotic potential of the chloroplast. Interaction of salinity and Trehalose improved photosynthetic pigment levels.
Trehalose is one of the compatible solutes, which are highly soluble, low molecular weight compounds which are usually non-toxic at high cellular concentrations. Generally, such solutes protect plants from stress through different courses, including contribution to cellular osmotic adjustment, detoxification of reactive oxygen species, protection of membrane integrity, and stabilization of enzymes and proteins.
Philosophy of this work is very good, applicable in basic and applied plant sciences. The innovation of the paper is strongly convected to the journal. The manuscript contains original data. The article is well organized. Material, Methods, and Protocols are standard. The experimental design was appropriate and the applied protocols were used correctly. The figures, captations content are considerable, readable and useful. The results of the experiments provide novel original data.
Specific comments:
Add new keywords – osmotic adjustment.
Response: Osmotic adjustment was added in keywords
What is the capacity for the accumulation of trehalose in cells.
Response:
Under in vivo conditions, trehalose has been shown to protect cells and organelles from denaturation, but only when present in high concentrations in the so-called anhydrobiotic organisms, such as yeast, tardigrades, and some plants, very high trehalose levels (above 10% of the dry weight) help these organisms to survive complete dehydration (Iturriaga et al., 2000).
for cryptobiotic species, such as the desiccation-tolerant S. lepidophylla. During its dehydration, trehalose accumulates to a level of 12% of the plant dry weight, and acts to protect proteins and membrane structures.
In higher vascular plants, accumulation of trehalose under adverse conditions is rare. It has been suggested that in most plant species sucrose has taken over the role of trehalose as a preservative during desiccation. However, in a few desiccation tolerant angiosperms trehalose is present in relatively large amounts. See line 41 and 42.
Does trehalose play a protective role against photoinhibition and high temperature similarly than proline?
Response: yes, trehalose play a protective role to tolerate photoinhibition and high temperature, please see line 73-79
Write more about the physiological significance of trehalose in plants. It was recently published about the improving effect of trehalose may be due to its physicochemical properties that stabilize dehydrated enzymes, proteins, and lipid membranes, as well as protect structures from damage during desiccation. Can trehalose improve the stability of antioxidative enzymes under stress?
Response: Ok, your comments were attending, please see introduction, see lines 48-58
What is the role of trehalose in alleviating in improving crop tolerance to oxidative stress induced by drought and salinity?
Response: Ok, we cited some research where discuss these roles of the trehalose to improving the abiotic stress tolerant. Please see line 73-79
Add characteristics of water stress during the experiment (RWC or leaf water potential) if it is possible.
Response: is not possible, we don’t evaluate this data.
Can be trehalose application recommended to farmers as an effective measure for the strengthen ecostability of physiological processes of plants under drought environments?
Response: Yes trehalose exogenous application can be effective to regulated physiological processes in plant under abiotic stress (Bae, et al 2005. Exogenous trehalose alters Arabidopsis transcripts involved in cell modification, abiotic stress, nitrogen metabolism and plant defense. Please see lines 55-58.
-Add more about osmotic adjustment. I would like to propose to improve introduction and discussion about the physiological significance of this kind of research, using some very important and innovative topical papers, e.g.:
* Abdelgawad ZA, Hathout TA, El-Khallal SM, Said EM, Al-Mokadem AZ.: Accumulation of trehalose mediates salt adaptation in rice seedlings. Am Eurasian J Agric Environ Sci 2014, 14(12):1450–1463
* Zhang D, Tong J, He X, et al.: A Novel Soybean Intrinsic Protein Gene, GmTIP2;3, Involved in Responding to Osmotic Stress. Front. Plant Sci. 2016, 6:1237. doi: 10.3389/fpls.2015.01237
* Mostafa MG, Hossain MA, Fujita M.: Trehalose pretreatment induces salt tolerance in rice (Oryza sativa L.) seedlings: oxidative damage and coinduction of antioxidant defense and glyoxalase systems. Protoplasma, 2015, 252: 461–475
* Ma C, Wang Z, Kong B, Lin T: Exogenous trehalose differentially modulate antioxidant defense system in wheat callus during water deficit and subsequent recovery. Plant Growth Regul., 2013, 70:275–285
Response: Thank you for your suggestion, the manuscripts were considerer for introduction and discussion.
Sincerely
Dra. Sugey Ramona Sinagawa-García
Corresponding author
Round 2
Reviewer 2 Report
Still the resubmitted MS fails to demonstrate that maize TPS and TPP respond to drought stress.
In fact, it is indicated at lines 29-31 that “The expression pattern showed significant induction after 0.5 hours in resistant lines and after one hour in susceptible plants”, at lines 230-231 that “The relative expression of both genes, TPS and TPP, was significantly induced after stress, showing their participation in drought stress response”, and at lines 307-308 that “we cloned the TPS and TPP genes from maize seedlings and found that these genes are differentially expressed under drought stress, showing their participation in drought stress response”, but Fig. 5 shows something different, i.e. that only minor variations in response to drought stress and only for a few of the times analyzed.
Therefore, The MS cannot be accepted.
Anyway, the problem remains Fig. 5 that need to be modified. I suggest to analyse the expression data of the two genes in seedlings drought stressed vs. unstressed seedlings creating two (one for TPS-3 and one for TPP-1) 4 parts figures showing: resistant seedlings unstressed, susceptible seedlings unstressed, resistant seedlings drought stressed, susceptible seedlings drought stressed. In this way significant differences in expression should be evident.
Author Response
Dear Ms. Hinata Fang
Assistant Editor Plants
Thank you for your attention for the revision of the manuscript entitled “Characterization of trehalose-6-phosphate synthase and trehalose-6-phosphate phosphatase genes and analysis of its differential expression in maize (Zea mays) seedlings under drought stress, (Manuscript ID: plants-701585) please find attached the revised version where we have attended all comments and suggestion of the reviewers in the manuscript.
The responses to reviewers’ comments are the following:
Academic editor comments
Even though the authors’ effort to respond to the reviewers’ comment and suggestion,
the revised manuscript has still several points to be addressed.
- The comments and suggestions of Reviewer 4 and 5 should be clearly addressed.
Response: The comments of the reviewer 4 and 5 were attended. Please see below
- Full discussion of the present data with two given references-below to advocate the novelty and significance of the present manuscript.
1. Wang, Song, Kai Ouyang, y Kai Wang. “Genome-Wide Identification, Evolution, and Expression Analysis of TPS and TPP Gene Families in Brachypodium distachyon. ”Plants 8, no.. 10 (September 23, 2019). 362.
2. Nature. 2006 May 11;441(7090):227-30. “A trehalose metabolic enzyme controls inflorescence architecture in maize". .http://doi.org/10.3390/plants8100362, is a genome-wide study.
Response: These manuscripts were considered and cited into introduction and discussion.
Reviewer 2
Anyway, the problem remains Fig. 5 that need to be modified. I suggest to analyses the expression data of the two genes in seedlings drought stressed vs. unstressed seedlings creating two (one for TPS-3 and one for TPP-1) 4 parts figures showing: resistant seedlings unstressed, susceptible seedlings unstressed, resistant seedlings drought stressed, susceptible seedlings drought stressed. In this way significant differences in expression should be evident.
Response: Now Materials and Methods described how the data of qRT-PCR were reported: “The relative fold change in gene expression was calculated using the 2 (-ΔΔCt) method, where, ΔCt = (Ct target gene - Ct reference gene), then the ΔCts were normalized using the Endogenous Control gene ΔΔCt = (ΔCt treated - ΔCt untreated) and finally the differential expression the TPS and TPP genes were calculated and represented in Log 2 scale”. Please see lines 341-344.
Fig. 5 now changed to Fig 6. Statistical analysis was added; as well as, the data showed was normalized using Log2 2(-ΔΔCt), where the endogenous control gene (well-watered fold change) take the value of 0. Thus, the level expression present was the fold change of the gene of interest based in stress treatment. The qPCR results were reported as Log2 2(-ΔΔCt). The statistically significant difference between the expression level was analyzed with a one-way ANOVA, and treatment means were compared with a Tukey Test (p ≤ 0.05) using Excel 2016 and SPSS statistical package version 25. See lines 345-348
Sincerely
Dra. Sugey Ramona Sinagawa-García
Corresponding author
Reviewer 3 Report
The reference the authors presented, Wang, Song, Kai Ouyang, y Kai Wang. “Genome-Wide Identification, Evolution, and Expression Analysis of TPS and TPP Gene Families in Brachypodium distachyon.”Plants 8, no.. 10 (September 23rd,2019): 362. https://doi.org/10.3390/plants8100362, is a genome-wide study. The paper the authors submitted is not a genome-wide study. The maize genome sequences are available in public databases, for example, https://phytozome.jgi.doe.gov/pz/portal.html.
Unfortunately, the genome-wide study of trehalose metabolic genes has been published and characterized in the previous paper, " Nature. 2006 May 11;441(7090):227-30. A trehalose metabolic enzyme controls inflorescence architecture in maize".
Therefore, in my view, it is ideally suited for publication in other journals.
Author Response
Dear Ms. Hinata Fang
Assistant Editor Plants
Thank you for your attention for the revision of the manuscript entitled “Characterization of trehalose-6-phosphate synthase and trehalose-6-phosphate phosphatase genes and analysis of its differential expression in maize (Zea mays) seedlings under drought stress, (Manuscript ID: plants-701585) please find attached the revised version where we have attended all comments and suggestion of the reviewers in the manuscript.
The responses to reviewers’ comments are the following:
Academic editor comments
Even though the authors’ effort to respond to the reviewers’ comment and suggestion,
the revised manuscript has still several points to be addressed.
- The comments and suggestions of Reviewer 4 and 5 should be clearly addressed.
Response: The comments of the reviewer 4 and 5 were attended. Please see below
- Full discussion of the present data with two given references-below to advocate the novelty and significance of the present manuscript.
1. Wang, Song, Kai Ouyang, y Kai Wang. “Genome-Wide Identification, Evolution, and Expression Analysis of TPS and TPP Gene Families in Brachypodium distachyon. ”Plants 8, no.. 10 (September 23, 2019). 362.
2. Nature. 2006 May 11;441(7090):227-30. “A trehalose metabolic enzyme controls inflorescence architecture in maize". .http://doi.org/10.3390/plants8100362, is a genome-wide study.
Response: These manuscripts were considered and cited into introduction and discussion.
Reviewer 3.
The reference the authors presented, Wang, Song, Kai Ouyang, y Kai Wang. “Genome-Wide Identification, Evolution, and Expression Analysis of TPS and TPP Gene Families in Brachypodium distachyon.”Plants 8, no.. 10 (September 23rd,2019): 362. https://doi.org/10.3390/plants8100362, is a genome-wide study. The paper the authors submitted is not a genome-wide study. The maize genome sequences are available in public databases, for example, https://phytozome.jgi.doe.gov/pz/portal.html.
Unfortunately, the genome-wide study of trehalose metabolic genes has been published and characterized in the previous paper, " Nature. 2006 May 11;441(7090):227-30. A trehalose metabolic enzyme controls inflorescence architecture in maize".
Therefore, in my view, it is ideally suited for publication in other journals.
Response: Thank you for your comments, the manuscript has presented several significant changes since its first version, now we see that two more reviewers are included; we believe that the comments of each of the reviewers, including the academic editor, have been very valuable for the improvement of this, which have been reflected in this version that we are sending. However, we think that you have the final decision.
Sincerely
Dra. Sugey Ramona Sinagawa-García
Corresponding author
Round 3
Reviewer 2 Report
I'm sorry, but the data shown in Figure 6 does not support lines 262-263: “The relative expression of both genes, TPS-3 and TPP-1, was significantly induced after stress, this behavior could be related to their participation in drought stress tolerance”. In fact, he trends of the expression for the two analyzed genes show only minimal differences between the resistant line and the sensitive line; moreover, drought stress doesn’t induce an upregulation of TPS-3 and leads to a clear upregulation for TPP-1 only after 18 hours.
Therefore, Authors must change their conclusions.
This manuscript is a resubmission of an earlier submission. The following is a list of the peer review reports and author responses from that submission.
Round 1
Reviewer 1 Report
This manuscript provides some information on TS and TPS from maize. The work is well conducted and described.
The role of this metabolism in stress response is considered but not the potential regulatory roles of these metabolites. Trehalose may also have a signalling role in plants. Literature searches should allow details to be added where appropriate. This would improve the overall value of the manuscript.
It would also benefit for some English language editing.
Reviewer 2 Report
The MS appears really confused and deserve rejection.
In fact, the data presented failed to demonstrate that maize TPS and TPP have a role in plant stress response, whereas so far, the mere characterisation of two genes it is not a condition for the publication of a scientific article.
Moreover, the quality of the manuscript is low; the main critical points are described below
No attempt to estimate trehalose contents (see e.g. reference 11); No reference to the enzyme trehalase which prevents accumulation of trehalose in higher plants; Line 432 (ref. 11), “Arabidopsis Thaliana” must be in Italics and with the lower case “t”: Arabidopsis thaliana; The abbreviation for the treatments / controls are difficult to understand; The leaves number is reported but apparently no indication of the DW of the seedlings after treatments; There is no clear indication for genes/proteins; e.g. do Zea mays (B4FVF6), maize (ZmTPP), ZmTPS (Zea mays, NP_001123593.2) and ZmTPP of (Zea Mays-B4FVF6) represent an identical gene / protein product? Does Figure1 show an average of the proline and SOD content between time 0,5 h and 72 h? I suggest a Figure including four lines; Figure 6; because the maize TPS genes are at least 3 and the TPP genes probably two, are the data presented an average or the relative expression of only one of the genes as indicate in the Figure legend? The reference 10 concerns trehalase overexpression not trehalose (production) genes.
Reviewer 3 Report
This manuscript deals with the characterization of the genes involved in trehalose biosynthesis in maize. I think the writing often lacks clarity and sharpness. This manuscript is not suitable for publication in this journal and should be submitted elsewhere. While I appreciate the effort of the work presented, I think the authors needs to improve the focus of the paper and provide more information on the significance of trehalose under drought conditions in maize.
1) Affiliations should be written in English.
2) Statistical analysis should be properly performed in Fig. 1 and Fig. 6.
3) The fonts in the figures are too small.